# Unraveling the Life Cycle of *Nyssopsora cedrelae*: A Study of Rust Diseases on *Aralia elata* and *Toona sinensis*

**DOI:** 10.3390/jof10040239

**Published:** 2024-03-22

**Authors:** Jae Sung Lee, Makoto Kakishima, Ji-Hyun Park, Hyeon-Dong Shin, Young-Joon Choi

**Affiliations:** 1Department of Biological Science, Kunsan National University, Gunsan 54150, Republic of Korea; jaesung1044@gmail.com; 2Faculty of Life and Environmental Sciences, University of Tsukuba, Tsukuba 305-8572, Japan; kakishima.makoto.ga@u.tsukuba.ac.jp; 3Department of Forestry, Environment and Systems, Kookmin University, Seoul 02707, Republic of Korea; jhpark10@kookmin.ac.kr; 4Division of Environmental Science and Ecological Engineering, College of Life Sciences and Biotechnology, Korea University, Seoul 02841, Republic of Korea; hdshin@korea.ac.kr; 5Center for Convergent Agrobioengineering, Kunsan National University, Gunsan 54150, Republic of Korea

**Keywords:** alternate host, Korean angelica tree, Pucciniales, *Puccinia caricis-araliae*

## Abstract

Rust disease poses a major threat to global agriculture and forestry. It is caused by types of Pucciniales, which often require alternate hosts for their life cycles. *Nyssopsora cedrelae* was previously identified as a rust pathogen on *Toona sinensis* in East and Southeast Asia. Although this species had been reported to be autoecious, completing its life cycle solely on *T. sinensis*, we hypothesized that it has a heteroecious life cycle, requiring an alternate host, since the spermogonial and aecial stages on *Aralia elata*, a plant native to East Asia, are frequently observed around the same area where *N. cedrelae* causes rust disease on *T. sinensis*. Upon collecting rust samples from both *A. elata* and *T. sinensis*, we confirmed that the rust species from both tree species exhibited matching internal transcribed spacer (ITS), large subunit (LSU) rDNA, and cytochrome oxidase subunit III (CO3) mtDNA sequences. Through cross-inoculations, we verified that aeciospores from *A. elata* produced a uredinial stage on *T. sinensis*. This study is the first report to clarify *A. elata* as an alternate host for *N. cedrelae*, thus providing initial evidence that the *Nyssopsora* species exhibits a heteroecious life cycle.

## 1. Introduction

Rust disease, caused by members of the order Pucciniales, poses a severe threat to diverse trees and crops [1,2]. This group often demonstrates a complex life cycle, which frequently involves switching between primary and alternate host plants. This heteroecious characteristic plays a central role in the epidemiology of rust diseases [3,4], requiring a comprehensive understanding of all host plants involved.

*Toona sinensis* (syn. *Cedrela sinensis*; Meliaceae), also known as the red toon or Chinese mahogany, is a tall deciduous tree that grows up to 20 m in height. This tree is native to East and Southeast Asia and is usually grown to produce high-quality timber, which is ideal for crafting furniture and musical instruments because of its sophisticated reddish colour and durability. In East Asia, its young leaves are utilized as a vegetable as well as for treating several ailments in traditional medicine [5,6].

*Aralia elata* (Araliaceae), also known as the Korean angelica tree, is a woody plant widely distributed throughout East Asia. This plant is often grown as an ornamental tree because of its unique characteristics, including spiny stems, toothed leaves, and clusters of small white flowers bearing black drupes as fruits. This tree is utilized as a traditional medicinal plant for its pharmacological effects, such as its anti-tumour, anti-inflammatory, and hepatoprotective effects [7]. In Korea, its young shoots are harvested in the spring and used in various Korean dishes because of their pleasant aroma and soft texture [8]. As of 2021, its production had reached 1383 tons, estimated to be worth 20 billion KRW [9].

*Nyssopsora cedrelae* is known as a rust pathogen that affects *Ailanthus altissima*, *Toona serrata* (=*Cedrela serrata*), and *T. sinensis* (=*C. sinensis*) [10], produces uredinial and telial stages, and has been reported in China, Japan, and Korea [11,12,13]. However, its life cycle is not fully understood. Through inoculation experiments using basidiospores obtained from teliospores, Kakishima et al. initially reported that this rust species could complete its life cycle on a single host [14], producing aecia (uredinoid aecia), uredinia, and telia entirely on *T. sinensis*. However, its aecial stage is not distinctly recognized in nature due to the morphological similarities between the aecial and uredinial stages, and no spermogonium has been found [11]. Consequently, the life cycle of *N. cedrelae* remains unclear.

Rust disease of *A. elata* is widespread throughout Korea. While the spermogonial and aecial stages of this rust have been observed on *A. elata*, the other stages of its life cycle remain elusive, leading to the conjecture that this rust may be heteroecious, utilizing different host plants for developing the other life stages. To date, two rust species, *Nyssopsora asiatica* and *Puccinia caricis-araliae* (also known as *Aecidium araliae*), have been reported on *A. elata* [11,12,15]. However, the morphologies and life cycles of these species are quite different. *N. asiatica* is a microcyclic autoecious species forming only telia on *Acanthopanax sciadophylloides*, *Aralia chinensis*, *A. cordata*, *A. elata*, *A. spinosa*, *Evodiopanax innovans*, *Kalopanax innovans*, and *Merrilliopanax listeri* [10,11]. The spermogonial and aecial stages that occur on *A. elata* in Korea are similar to those that occur on *P. caricis-araliae* in their symptoms but differ in morphology, and these stages are frequently observed in areas where *N. cedrelae* occurs on *T. sinensis*. Therefore, we suspected that these stages on *A. elata* are in fact the spermogonial and aecial stages of *N. cedrelae*.

This study is the first to report *A. elata* as an alternate host for *N. cedrelae*, thus providing initial evidence that the *Nyssopsora* species exhibits a heteroecious life cycle. In the present study, we comprehensively characterized rust disease samples on *A. elata* and *T. sinensis* through morphological and molecular phylogenetic analyses as well as cross-inoculation tests. We aimed to identify the causal agent of rust disease on *A. elata* in Korea and to clarify the relationships of its spermogonial and aecial stages on *A. elata* with the uredinial and telial stages of *N. cedrelae* on *T. sinensis.*

## 2. Materials and Methods

### 2.1. Sample Collection

Thirty-three rust samples from *Aralia elata* and *Toona sinensis* were collected across various locations in Korea. Rust-infected leaves were prepared as dried specimens and preserved at the Kunsan National University (KSNUH) and Korea University (KUS-F) for further processing. In addition, three Japanese specimens of *N. cedrelae* were provided by the herbarium of the Department of Botany, National Museum of Nature and Science, Tsukuba, Japan (TNS-F), for comparison with the Korean samples. All herbarium specimens used for molecular phylogenetic and morphological analyses in this study are summarized in Table 1.

### 2.2. DNA Extraction, Amplification, Sequencing and Phylogenetic Analysis

Genomic DNA was extracted from rust-infected samples using a MagListo 5M Plant Genomic DNA Extraction Kit (Bioneer, Daejeon, Republic of Korea), following the manufacturer’s instructions. Polymerase chain reaction (PCR) was performed to amplify the internal transcribed spacer (ITS) rDNA region with primers ITS5u [16] and ITS4rust [17], large subunit (LSU) regions with primers LRust1R and LRust3 [16], and the cytochrome oxidase subunit III (CO3) mtDNA region with primers CO3-F1 and CO3-R1 [16]. The PCR products were purified using an AccuPrep^®^ PCR/Gel Purification Kit (Bioneer, Daejeon, Republic of Korea) and sequenced by the Macrogen sequencing service (Macrogen, Seoul, Republic of Korea). The resulting sequences were edited using DNASTAR software 7.1 (Lasergen, Madison, WI, USA).

The ITS, LSU, and CO3 sequences were compared to those of the closest related species in the GenBank database using the Basic Local Alignment Search Tool (BLASTn). The sequences of each marker were aligned using the FFT-NS-2 algorithm method in MAFFT version 7 [18]. Phylogenetic trees were constructed using the minimum evolution and maximum likelihood methods based on the Tamura–Nei model in MEGA 7 [19]. Statistical support for the branches of the phylogenetic trees was evaluated by the bootstrap method with 1000 replicates. Reference sequences from GenBank used in the phylogenetic analysis are listed in Table 2.

### 2.3. Morphological Analysis

The symptoms and macrostructures of rust-infected specimens were observed under a stereomicroscope (M205C; Leica, Wetzlar, Germany). The micromorphological characteristics were examined and photographed using a differential interference contrast (DIC) light microscope (Axio Imager 2; Carl Zeiss, Oberkochen, Germany). At least 50 rust sori and spores were measured per sample, and their measurements are represented as follows: (minimum–) standard deviation towards the minimum—standard deviation towards the maximum (–maximum) (mean). Scanning electron microscopy (SEM) (S-4800+EDS; Hitachi, Tokyo, Japan) was used for detailed morphological analysis.

### 2.4. Cross-Inoculation Experiments

Cross-inoculation experiments were conducted to demonstrate the pathogenicity of aeciospores from *A. elata* on *T. sinensis*. Aeciospores from rust-infected leaves of *A. elata* (KSNUH1831) were harvested using a spore collector (Tallgrass Solutions Inc., Manhattan, NY, USA) and stored in a refrigerator at 4 °C for an hour. Three healthy *T. sinensis* plants were inoculated by spraying a suspension of aeciospores in sterile water (1.1 × 10^6^) onto their leaves. Inoculated plants were then kept in a humid chamber at room temperature (25 °C) for three weeks and monitored for rust-symptom development. Two non-inoculated plants served as controls.

## 3. Results

### 3.1. Phylogeny

The ITS and LSU rDNA sequences of rust samples collected from *A. elata* and *T. sinensis* were identical. Among the 33 rust samples, slight sequence differences were observed at two sites in the ITS region and one site in the LSU region. BLASTn searches revealed that the Korean and Japanese samples were closest to *Nyssopsora altissima* from *Ailanthus altissima* in China. However, there were 17 nucleotide differences from *N. altissima* in the ITS sequences and a gap in the LSU sequences. In the phylogenetic trees of the concatenated alignment of ITS and LSU sequences (Figure 1), samples from both host plants were consistently grouped with the maximum bootstrapping support value, indicating the robustness of this phylogenetic grouping. The phylogenetic trees revealed two distinct clades within the *Nyssopsora* species based on their host plants. A clade that includes *N. cedrelae* shares the same host plants, *A. elata* and *T. sinensis*, whereas members of the other clade, including *N. echinata* (type species of *Nyssopsora*), originated from various host plants.

The CO3 sequences, spanning 649 bp, exhibited no sequence differences across all rust samples. In the phylogenetic tree of the CO3 sequences (Figure 2), samples from both host plants formed a distinct clade that had the highest level of bootstrapping support. Moreover, this clade was distinctly segregated from the Gymnosporangiaceae, Pucciniaceae, and Sphaerophragmiaceae families, further underscoring the unique phylogenetic position of our samples.

### 3.2. Morphology

The symptoms of the Korean rust specimens on *A. elata* appeared as chlorotic spots, forming spermatogonia and aecia (Figure 3A–F). The infected leaves and stems became deformed (Figure 3C), and as the disease progressed, they increasingly withered (Figure 3D). The spermogonia were epiphyllous, scattered, subepidermal, yellow, and conical-shaped (type 5 of Cummins and Hiratsuka [2]), and measured 100–200 μm in diameter (Figure 3G). The aecia were hypophyllous or cauligenous, yellow to orange, cupulate with peridia, and measured 450–1250 μm (av. 770 μm) in diameter (Figure 3E,F,J). Peridial cells were rectangular, rhomboid, and measured (16.5–)19.6–23.0(–25.7) × (10.4–)12.7–16.6(–20.5) μm (av. 21.36 × 14.78 μm), with a thick verrucose wall (Figure 3H). Aeciospores were globose to subglobose, pale yellow, and measured (14.1–)14.4–16.5(–18.9) × (11.9–)12.5–14.6(–17.0) μm (av. 15.45 × 13.61 μm), with a verrucose, thin wall containing 1–2 large granules (Figure 3I,K,L).

Rust symptoms on *T. sinensis* appeared as chlorotic spots on the upper leaf surface, forming uredinia and telia on the lower leaf surface (Figure 4A–F). During the uredinial stage, infected leaves exhibited light green chlorotic spots on the upper surfaces (Figure 4C). As the disease progressed to the telial stage, the leaves dried progressively and shed prematurely (Figure 4E). Uredinia on the *T. sinensis* collected in Korea were amphigenous, mostly hypophyllous, erumpent, scattered or aggregated, yellow to orange, round, and measured 200–589 μm (av. 354 μm) in diameter (Figure 4D,G,K). Urediniospores were mostly subglobose, rarely obovoid, yellowish, and measured (15.4–)16.6–19.1(–21.4) × (12.2–)14.9–17.2(–18.8) μm (av. 17.88 × 16.06), with an echinulate wall 1.5–3.0 μm in thickness (Figure 4H,L; Table 2). Telia were amphigenous, mostly hypophyllous, erumpent, scattered or aggregated, dark brown to black, pulverulent, and measured 500–2400 μm (av. 1123 μm) in diameter (Figure 4F,I,M). Teliospores were 3-celled, subglobose-trigonal, dark brown, with a hyaline pedicel on each spore, and measured (28.2–)30.2–33.6(–35.6) × (28.6–)30.4–34.9(–36.8) (av. 31.61 × 32.71 μm) (Figure 4J). The walls were smooth, 1.0–2.5 μm thick, light brown, with 13–21 projections on each spore, bi- or tri-branched tips, and measured 3.0–9.5 μm (Figure 4N).

### 3.3. Pathogenicity

When healthy *T. sinensis* leaves were inoculated with aeciospores from *A. elata* (Figure 5A,B as a control), chlorotic spots began to appear on the leaf surfaces two weeks after inoculation (Figure 5C). The symptoms were similar to those observed in the natural environment. After three weeks, all inoculated plants exhibited more pronounced rust symptoms and formed yellow uredinia on their leaf surfaces (Figure 5D–G), from which echinulate urediniospores were produced (Figure 5H), matching the morphological features of *N. cedrelae*. After five weeks, three of these plants persisted in the uredinial stage without progressing to the telial stage in the experiment.

## 4. Discussion

In the present study, we uncovered the life cycle of the rust pathogen *N. cedrelae*. Morphologically, the rust samples on *Aralia elata*, an essential woody plant in Korean cuisine, were somewhat similar to the characteristics of those from *Puccinia caricis-araliae* [15], another rust species found on *A. elata*. They had unique large granules, but the aeciospores were smaller than those of *P. caricis-araliae* (15.45 × 13.61 μm in *Nyssopsora cedrelae* versus 21.0 × 18.5 μm in *P. caricis-araliae*). Further, the type of spermogonia differed between the Korean samples and *P. caricis-araliae* (type 5 versus type 4). The characteristics of the projections on the teliospore walls of the *T. sinensis* samples corresponded well with those of *N. cedrelae* rather than of other *Nyssopsora* species, even though the teliospores from the present Korean and Japanese samples were smaller than those previously described (Table 3). The features of urediniospores closely matched those of *N. cedrelae*. Our phylogenetic study supported the notion that *N. cedrelae* is a rust pathogen affecting *Aralia elata* in Korea as well as *Toona sinensis* in Japan and Korea. Although our samples were morphologically similar to *Nyssopsora altissima* which has been described from *Ailanthus altissima* in China [20], they exhibited many sequence differences in the ITS regions.

As rust diseases pose a significant risk to forestry and agricultural productivity due to their severe impact on crop yield and quality [22,23], understanding their life cycles, especially their alternate hosts, can potentially enhance disease management strategies [3,4]. Our study revealed that *N. cedrelae*, a rust pathogen associated with *T. sinensis*, has an alternate host, namely, *A. elata*. The molecular phylogenetic identity found in the rust species affecting *A. elata* and *T. sinensis* provides substantial evidence linking the rust diseases on the two trees. The results of the inoculation test demonstrated that *A. elata* is a spermogonial and aecial host (alternate host) of *N. cedrelae*, thereby confirming the heteroecious life cycle of this rust pathogen. Kakishima et al. reported its autoecious life cycle, producing aecia (uredinoid aecia), uredinia, and telia, with basidiospore inoculations obtained from teliospores [14]. However, this result may be due to inoculum contamination with urediniospores during basidiospore inoculations because spermogonia were not reported in the inoculations, and uredinoid aecia were produced after basidiospore inoculations. These uredinoid aecia are suspected to present as uredinia after infection with urediniospores. Our results resolve the long-standing enigma that is the life cycle of *N. cedrelae,* contributing to a better understanding of the epidemiology and dispersion of this pathogen.

Globally, thirteen *Nyssopsora* species have been reported on various woody plants, including Anacardiaceae, Apiaceae, Araliaceae, and Meliaceae [2,11,24,25,26]. To date, their aecial stage has not been observed, leading to the speculation that they exhibit an autoecious life cycle, either microcyclic (producing only the telial stage) or hemicyclic (producing the uredial and telial stages). Our results indicate the potential presence of alternate hosts in the life cycle of the genus *Nyssopsora* and provide compelling evidence that supports the hypothesis of Henderson [27] that some *Nyssopsora* species, including *N. cedrelae* and *N. koelreuteriae*, might exhibit a heteroecious life cycle by producing an aecial stage on Apiaceae or Araliaceae. This finding represents not only the first observation of the spermogonial and aecial stages but also the first report of host-alternating in the family Nyssopsoraceae. Our results highlight the benefits of integrating traditional cross-inoculation testing with advanced molecular methods for studying rust pathogens and their complex life cycles.

## 5. Conclusions

This study represents a substantial advancement in our understanding of the dynamics of rust diseases affecting two economically valuable trees, *A. elata* and *T. sinensis*. We revealed the widespread presence of *N. cedrelae* on *A. elata* and elucidated its heteroecious life cycle, alternating between *A. elata* and *T. sinensis*. This finding emphasizes the potential threat that *N. cedrelae* poses to the cultivation and economic value of these two species. The insights gained from the current research are crucial for developing efficient approaches for managing rust diseases on these trees.

## Figures and Tables

**Figure 1 jof-10-00239-f001:**
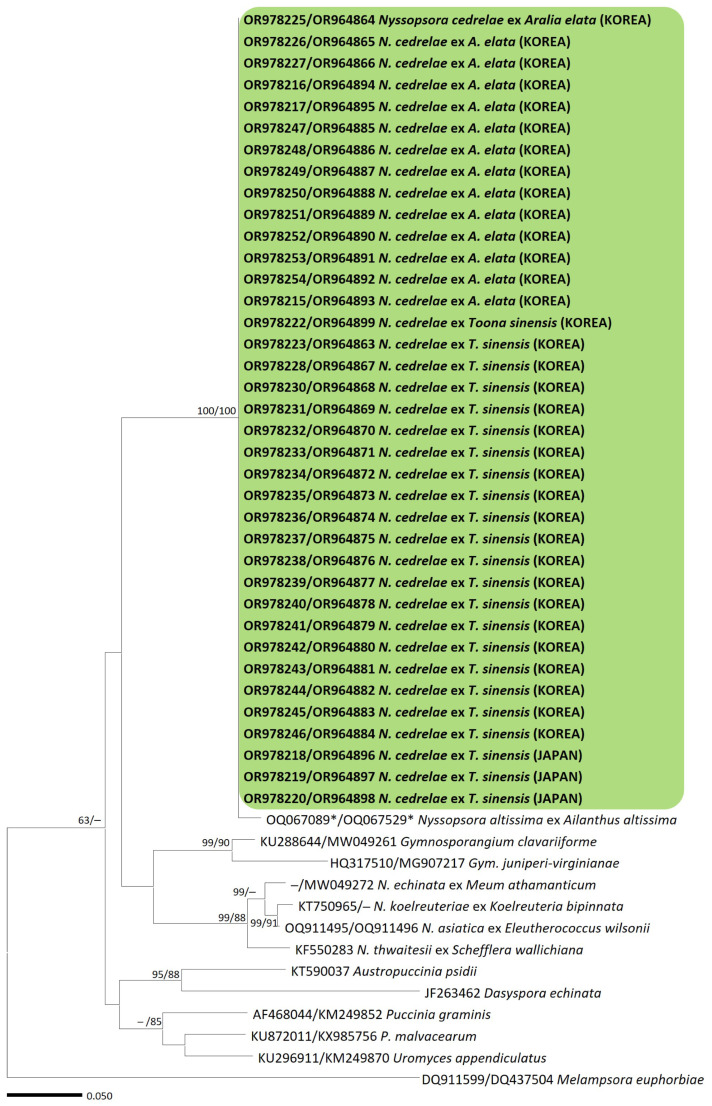
Maximum likelihood trees of rust species based on a concatenated alignment of the internal transcribed spacer (ITS) and large subunit (LSU) sequences. Bootstrapping support values (minimum evolution/maximum likelihood) higher than 60% are given above or below the branches. The clade, including *Nyssopsora cedrelae*, is highlighted in a green box, and the rust samples sequenced in this study are shown in bold. Asterisks (*) indicate sequences of holotype.

**Figure 2 jof-10-00239-f002:**
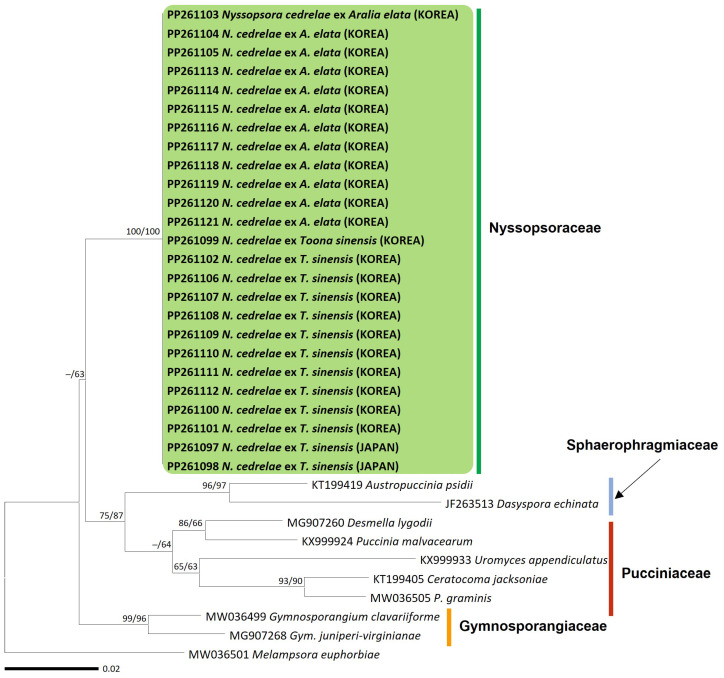
Maximum likelihood trees of rust species based on cytochrome oxidase subunit III (CO3) rDNA sequences. Bootstrapping support values (minimum evolution/maximum likelihood) higher than 60% are given above the branches. The clade, including *Nyssopsora cedrelae*, is highlighted in a green box, and the rust samples sequenced in this study are shown in bold.

**Figure 3 jof-10-00239-f003:**
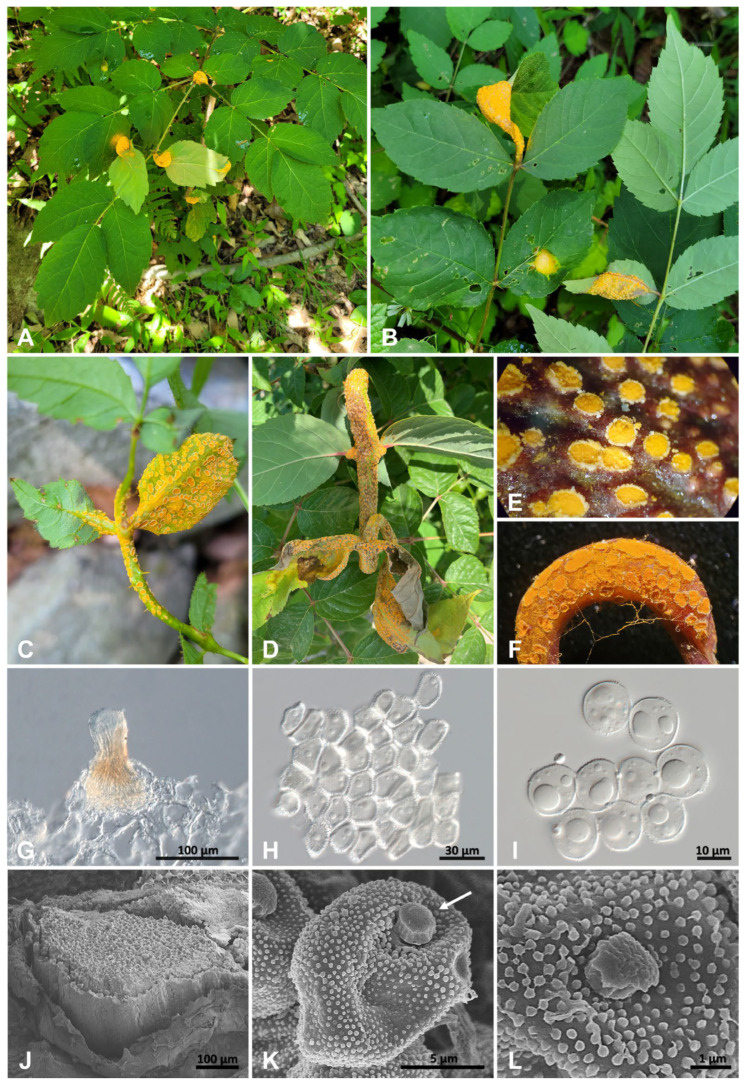
Rust disease on *Aralia elata* caused by *Nyssopsora cedrelae*. (**A**,**B**) Infected leaves of *A. elata*. (**C**) Deformed leaf and stem caused by rust infection. (**D**) Withered leaves. (**E**) Aecia in the early stage of the disease. (**F**) Aecia in the later stages of disease. (**G**) Spermogonium. (**H**) Peridial cells. (**I**) Aeciospores. (**J**) Aecium. (**K**) Aeciospore with a granule (arrow). (**L**) Verrucous wall ornamentation of aeciospore.

**Figure 4 jof-10-00239-f004:**
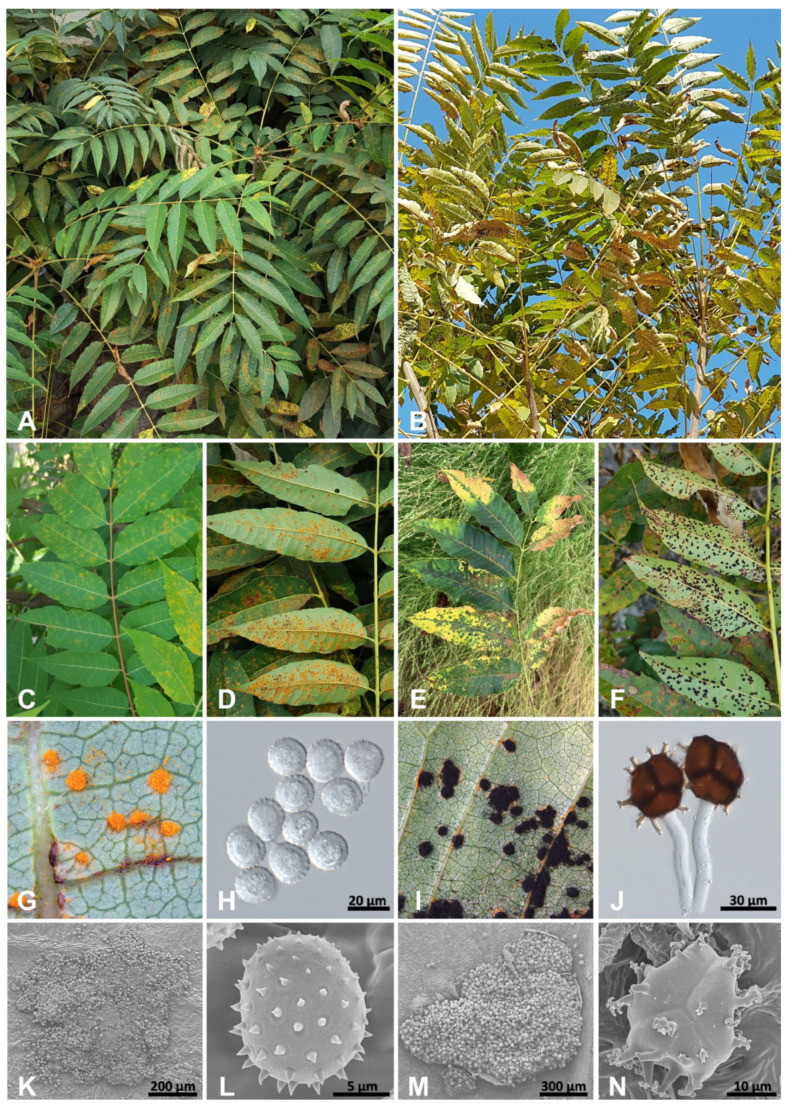
Rust disease on Toona sinensis caused by Nyssopsora cedrelae. (**A**,**B**) Infected leaves of T. sinensis. (**C**) The chlorotic spots on the upper leaf surface in uredinial stage. (**D**) Uredinia on the lower leaf surface. (**E**) The chlorotic spots on the upper leaf surface in telial stage. (**F**) Telia on the lower leaf surface. (**G**) Uredinia. (**H**) Urediniospores. (**I**) Telia. (**J**) Teliospores. (**K**) Uredinium. (**L**) Urediniospore. (**M**) Telium. (**N**) Teliospore.

**Figure 5 jof-10-00239-f005:**
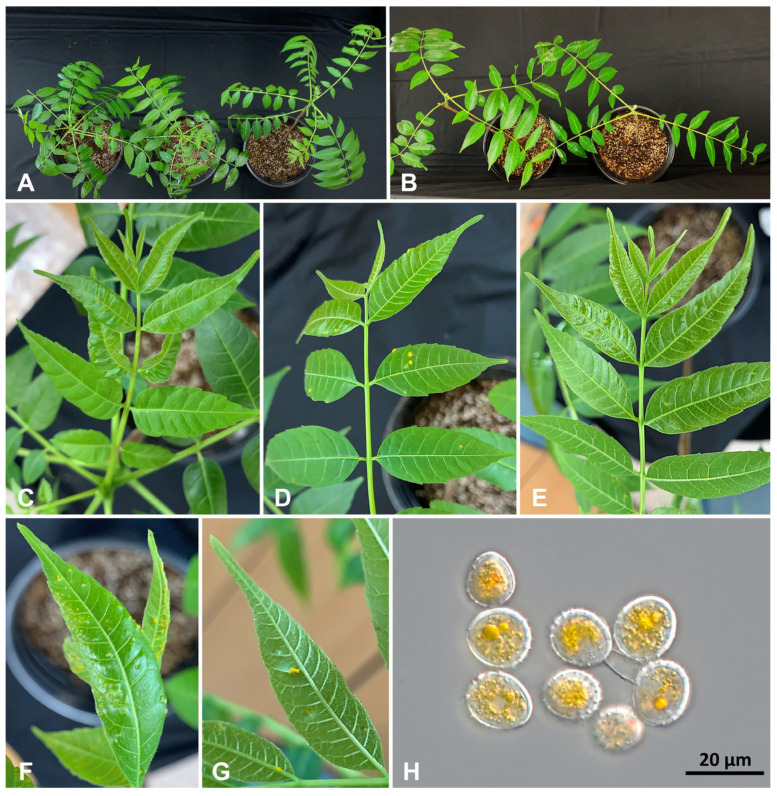
Cross-inoculation test. (**A**) Inoculation of aeciospores from *Aralia elata* on *Toona sinensis* leaves. (**B**) Controls. (**C**) the leaves with chlorotic spots two weeks after inoculation. (**D**,**E**) the rust symptoms on the upper (**D**) and lower (**E**) surfaces of infected leaves three weeks after inoculation. (**F**,**G**) Uredinia on infected leaves by inoculation. (**H**) Urediniospores from emerging uredinia.

**Table 1 jof-10-00239-t001:** Rust specimens of *Aralia elata* and *Toona sinensis* used in the present study.

Host Plant	Specimens No. *	Stage **	Locality	Collection Date	GenBank Acc. No.		
					ITS	LSU	CO3
*Aralia elata*	KSNUH1831	S, A	Korea, Namwon-si, Jusaeng-myeon	4 June 2022	OR978225	OR964864	PP261103
	KSNUH1848	S, A	Korea, Wanju-gun, Gosan Arboretum	25 May 2022	OR978226	OR964865	PP261104
	KSNUH1926	S, A	Korea, Jeonju-si, Wansan-gu, Jungin-dong	10 June 2022	OR978227	OR964866	PP261105
	KUS-F12861	S, A	Korea, Gangneung-si, Jibyeon-dong	8 June 1994	—	—	—
	KUS-F13838	S, A	Korea, Samcheok-si, Miro-myeon	26 May 1997	—	—	—
	KUS-F16018	S, A	Korea, Seoul, National Insitute of Forest Science	17 June 1999	—	—	—
	KUS-F24142	S, A	Korea, Bonghwa-gun, Chunyang-myeon, Seobyeok-ri	14 June 2009	—	—	—
	KUS-F30646	S, A	Korea, Pyeongchang-gun, Jinbu-myeon, Songjeong-ri	20 June 2018	OR978216	OR964894	—
	KUS-F31045	S, A	Korea, Yangpyeong-gun, Okcheon-myeon, Jungmisan Recreation Forest	24 June 2019	OR978217	OR964895	—
	KUS-F32831	S, A	Korea, Suncheon-si, Gangcheonsan County Park	18 May 2022	OR978247	OR964885	PP261113
	KUS-F32847	S, A	Korea, Jeonju-si, Mt. Geonjisan	22 May 2022	OR978248	OR964886	PP261114
	KUS-F32861	S, A	Korea, Wanju-gun, Soyang Eco-Forest	26 May 2022	OR978249	OR964887	PP261115
	KUS-F32865	S, A	Korea, Imsil-gun, Mt. Seongsusan	27 May 2022	OR978250	OR964888	PP261116
	KUS-F32875	S, A	Korea, Jangsu-gun, Seonggwansa Temple	30 May 2022	OR978251	OR964889	PP261117
	KUS-F32913	S, A	Korea, Wanju-gun, Daea Arboretum	3 June 2022	OR978252	OR964890	PP261118
	KUS-F32921	S, A	Korea, Wanju-gun, Pyeonbaek Forest	7 June 2022	OR978253	OR964891	PP261119
	KUS-F33693	S, A	Korea, Wanju-gun, Wibong Mountain Fortress	29 May 2023	OR978254	OR964892	PP261120
	KUS-F33702	S, A	Korea, Buan-gun, Sangseo-myeon, Gaeamsa Temple	30 May 2023	OR978215	OR964893	PP261121
*Toona sinensis*	KSNUH0876	U	Korea, Gunsan-si, Oksan-myeon, Oksan-ri	16 September 2020	OR978221	—	PP261099
	KSNUH1478	U	Korea, Chilgok-gun, Gisan-myeon, Yeong-ri	14 July 2021	OR978222	OR964899	PP261100
	KSNUH1519	U	Korea, Gunsan-si, Miryong-dong	5 August 2021	OR978223	OR964863	PP261101
	KSNUH1651	T	Korea, Yeongdong-gun, Hwanggan-myeon, Masan-ri	16 October 2021	OR978224	—	PP261102
	KSNUH1957	U	Korea, Buan-gun, Sangseo-myeon, Gamgyo-ri,	8 July 2022	OR978228	OR964867	PP261106
	KSNUH1981	T	Korea, Iksan-si, Osan-myeon, Jangsin-ri	28 September 2022	OR978229	—	—
	KUS-F17635	T	Korea, Cheongju-si, Chungcheongbuk-do Agricultural Research and Extension Services	24 September 2000	OR978230	OR964868	—
	KUS-F18055	T	Korea, Seoul, Forest Research Institute	6 November 2000	OR978231	OR964869	—
	KUS-F19928	T	Korea, Jinju-si, Geumsan-myeon, Cheonggoksa Temple	16 October 2003	OR978232	OR964870	—
	KUS-F23179	U	Korea, Suwon-si, Seodun-dong	7 November 2007	—	—	—
	KUS-F23191	T	Korea, Samcheok-si, Jukseoru Pavilion	9 November 2007	OR978233	OR964871	PP261107
	KUS-F23988	T	Korea, Gimhae-si, Daedong-myeon	21 November 2008	OR978234	OR964872	PP261108
	KUS-F24567	U, T	Korea, Yangyang-gun, Hyeonbuk-myeon	14 September 2009	OR978235	OR964873	PP261109
	KUS-F25499	U	Korea, Pocheon-si, Korea National Arboretum	18 October 2010	—	—	—
	KUS-F25500	T	Korea, Pocheon-si, Korea National Arboretum	18 October 2010	OR978236	OR964874	—
	KUS-F27646	U	Korea, Milyang-si, Mt. Yongdusan	26 September 2013	OR978237	OR964875	—
	KUS-F29339	U	Korea, Wando-gun, Wando Arboretum	26 July 2016	OR978238	OR964876	—
	KUS-F29353	U	Korea, Iksan-si, Jeollabuk-do Agricultural Research and Extension Services	3 August 2016	OR978239	OR964877	—
	KUS-F29959	U	Korea, Gimcheon-si, Mt. Geumosan	7 September 2017	OR978240	OR964878	PP261110
	KUS-F30343	T	Korea, Gimcheon-si, Geumo-dong	7 November 2017	OR978241	OR964879	—
	KUS-F31343	U, T	Korea, Gimcheon-si, Yogok-dong	21 October 2019	OR978242	OR964880	—
	KUS-F31776	U	Korea, Gimcheon-si, Nam-myeon, Mt. Unnamsan	23 June 2020	OR978243	OR964881	—
	KUS-F32487	T	Korea, Wanju-gun, Soyang-myeon, Wibongsa, Temple	1 October 2021	OR978244	OR964882	—
	KUS-F32982	U	Korea, Buan-gun, Sangseo-myeon, Gaeamsa Temple	20 June 2022	OR978245	OR964883	PP261111
	KUS-F33532	T	Korea, Gochang-gun, Haeri-myeon, Songyangsa Temple	10 November 2022	OR978246	OR964884	PP261112
	TNS-F99270	U	Japan, Chiba, Noda	16 October 2022	OR978218	OR964896	PP261097
	TNS-F99271	T	Japan, Ibaraki, Inashiki	14 November 2022	OR978219	OR964897	—
	TNS-F99272	T	Japan Ibaraki, Ushiku	20 November 2022	OR978220	OR964898	PP261098

* KSNUH: Kunsan National University Herbarium; KUS-F: Fungal specimens of Korea University; TNS-F: Fungal specimens of National Museum of Nature and Science. ** S = spermogonial stage; A = aecial stage; U = uredinial stage; T = telial stage.

**Table 2 jof-10-00239-t002:** List of reference sequences of Nyssopsoraceae used for phylogenetic analysis.

Rust Species	Host Plants	Specimens *	GenBank Accession No.		
			ITS	LSU	CO3
*Austropuccinia psidii*	*Melaleuca leucadendra*	PREM 61282	KT590037	KT590037	—
	*Rhodamnia angustifolia*	BRIP 57793	—	—	KT199419
*Ceratocoma jacksoniae*	*Daviesia* sp.	BRIP 57762	—	—	KT199405
*Dasyspora echinata*	*Xylopia aromatica*	BPI 746651	JF263462	JF263462	—
	*Xylopia aromatica*	ZT HeRB8486	—	—	JF263513
*Desmella lygodii*	*Lygodium japonicum*	U1226	—	—	MG907260
*Gymnosporangium clavariiforme*	*Malus* sp.	HMAS:67951	KU288644	—	—
	*Crataegus* sp.	BRIP 59471	—	MW049261	MW036499
*Gym. juniperi-virginianae*	*Juniperus virginiana*	DAOM 234434	HQ317510	—	—
	*Juniperus* sp.	MCA3585	—	MG907217	MG907268
*Melampsora euphorbiae*	No data	U-00138	DQ911599	—	—
	*Euphorbia macroclada*	BPI 863501	—	DQ437504	—
	*Euphorbia macroclada*	BPI 86350	—	—	MW036501
*Nyssopsora altissima*	*Ailanthus altissimus*	GMB0103 **	OQ067089	OQ067529	—
*N. asiatica*	*Eleutherococcus wilsonii*	QHU2022221	OQ911496	OQ911495	—
*N. echinata*	*Meum athamanticum*	KR0012164	—	MW049272	—
*N. koelreuteriae*	*Koelreuteria bipinnata*	BBSW-1	KT750965	—	—
*N. thwaitesii*	*Schefflera wallichiana*	AMH 9528	KF550283	KF550283	—
*Puccinia graminis*	*Agropyron repens*	No data	AF468044	—	—
	*Glyceria maxima*	BRIP:60137	—	KM249852	MW036505
*Puccinia malvacearum*	*Malva sylvestris*	INU_12572-2016	KU872011	—	
	*Malva nicaeensis*	PDD:101511	—	KX985756	—
	*Malva parviflora*	BRIP 57522	—	—	KX999924
*Uromyces appendiculatus*	*Macroptilium atropurpureum*	BRIP 60929	KU296911	—	—
	*Phaseolus vulgaris*	BRIP 60020	—	KM249870	KX999933

* AMH: Agharkar Research Institute, India; BBSW: BeiBei Southwest University, China; BPI: U.S. National Fungus Collections, USDA-ARS, USA.; BRIP: Queensland Plant Pathology Herbarium, Australia; DAOM: Canadian National Mycological Herbarium-AAFC, Canada; GMB: Guizhou Medical University, China; HMAS: Chinese Academy of Sciences, China; INU: Inonu University, Turkey; KR: Staatliches Museum für Naturkunde Karlsruhe; MCA: Muhlenberg College, USA.; PDD: Manaaki Whenua—Landcare Research, New Zealand; PREM: Plant Protection Research Institute, Republic of South Africa; QHU: Qinghai University, China; U: Naturalis Biodiversity Center, The Netherlands; ZT: Herbarium der Eidgenössische Technische Hochschule Zürich. ** A-type specimen (holotype).

**Table 3 jof-10-00239-t003:** Morphological characteristics of *Nyssopsora cedrelae* on *Toona sinensis*.

Reference	Country	Urediniospores	Teliospores			Specimen
		Size (µm)	Size (µm)	Projections		
				No.	Length (µ)	
Lohsomboon, Kakishima [11]	Japan	14–24 × 13–21 (av. 18–16)	29–44 × 27–44 (av. 35–34)	13–27	3–9	PUR-68828 (isotype)
Hori [21]	Japan	17–21	35–40	–	7	- ^a^
Lütjeharms [10]	–	–	–	20–30 (av. 26)	–	Nr. 44
In this study	Korea	15.4–21.4 × 12.2–18.8 (av. 17.88 × 16.06)	–	–	–	KSNUH1478
	Korea	–	25.1–34.4 × 16.6–35.5 (av. 30.03 × 31.79)	13–22	5.3–6.9	KSNUH1651
	Korea	16.7–20.2 × 15.1–18.7 (av. 18.14 ×16.91)	28.2–35.6 × 28.6–36.8 (av. 31.61 × 32.71)	14–21	3.2–8.3	KUS-F23988
	Korea	–	25.2–33.0 × 25.9–34.2 (av. 30.29 × 30.08)	15–23	2.9–6.8	KUS-F25500
	Korea	14.4–20.1 × 12.8–17.1 (av. 16.80 × 15.09)	25.1–34.4 × 16.6–35.5 (av. 30.59 × 30.97)	14–24	4.1–9.5	KUS-F31343
	Japan	15.0–17.9 × 13.0–16.2 (av. 16.40 × 15.21)	–	–	–	TNS-F99270
	Japan	–	26.8–33.8 × 25–34.7 (av. 30.38 × 31.12)	15–22	3.1–7.5	TNS-F99271
	Japan	–	25.0–35.5 × 27–35.8 (av. 31.18 × 31.02)	14–18	3.3–9.4	TNS-F99272

^a^ The morphological characteristics were measured from a specimen collected on *Cedrela sinensis* (now *Toona sinensis*) (Botanical Garden, Tokyo, Japan, 15 October 1891, S. Hori), without a herbarium number.

## Data Availability

The data will be made available by the authors on request.

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
