# Peer review of "Unraveling the Life Cycle of Nyssopsora cedrelae: A Study of Rust Diseases on Aralia elata and Toona sinensis"

_jof, 2024, doi:10.3390/jof10040239_

Round 1

Reviewer 1 Report

Comments and Suggestions for Authors

The study of the life cycle of rust fungi is of great importance and is currently a field of scientific research due to its lack of study. The authors have conducted more exciting research. research, but there are still areas that require improvement. For instance, the suggested modification areas are as follows:

1. Validation of phylogenetic relationships requires supplementation of results from other molecular marker tests, despite the completion of ITS- and LSU-based sequence analyses.

2. Molecular characterization is necessary to determine the phylogenetic relationships among the 35 strains (OR964864-OR964898) within the same species of N. cedrelae.

3. Clarification is needed regarding which of the 35 strains was selected for the inoculation test. In addition, a theoretical basis for molecular biology and pathology should be provided.

4. Additional inoculation tests using basidiospores should be conducted.

5. Additional micrographs of rust spore inoculation and photographs taken at different time points during pathogenesis after inoculation are necessary.

The list of recommendations provided above is not exhaustive. The authors are encouraged to conduct additional experiments to further enhance this paper.

Author Response

  1. Validation of phylogenetic relationships requires supplementation of results from other molecular marker tests, despite the completion of ITS- and LSU-based sequence analyses.

→ In this revision, the authors have added the CO3 (cytochrome oxidase 3) mtDNA marker, with the resulting phylogenetic tree presented in Figure 2.

  1. Molecular characterization is necessary to determine the phylogenetic relationships among the 35 strains (OR964864-OR964898) within the same species of N. cedrelae.

→ The authors have detailed sequence differences among the 35 strains at Lines 129-130.

  1. Clarification is needed regarding which of the 35 strains was selected for the inoculation test. In addition, a theoretical basis for molecular biology and pathology should be provided.

→ The authors have specified the strain (specimen number) used as inoculum for the pathogenicity assay at Line 120.

  1. Additional inoculation tests using basidiospores should be conducted.

→ Reciprocal inoculation is crucial for confirming the life cycle. However, conducting both aeciospore and basidiospore inoculations can be challenging due to the difficulty of obtaining suitable materials. Thus, a one-way inoculation has been utilized to substantiate the life cycle.

  1. Additional micrographs of rust spore inoculation and photographs taken at different time points during pathogenesis after inoculation are necessary.

→ The authors believe that the photographs provided sufficiently demonstrate the results of the inoculations, as the focus of this study is not on the pathogenetic process.

The list of recommendations provided above is not exhaustive. The authors are encouraged to conduct additional experiments to further enhance this paper.

→ All suggestions from Reviewer 1 have been addressed, with the exception of those pertaining to additional inoculation tests.

Reviewer 2 Report

Comments and Suggestions for Authors

This article is about identifying the phenomenon of rust disease transmission to alternate hosts, combining morphological, molecular biology, and inoculation experiments. The manuscript is carefully prepared and presented. The authors ultimately summarized the potential life cycle of Nyssopsora cedrelae, which includes the Spermogonial and Aecial stages on Aralia elata and the Uredinial and Telial stages on Toona sinensis. However, some parts need to be revised.

1. In the background introduction section, more discussion about the pathogen is needed, such as its taxonomic status and pathogenicity.

2. The author also conducted inoculation experiments on a series of potential host plants to determine the host range and potential for host switching of the rust fungus.

3. Molecular marker analysis can be used to assess the genetic diversity and structure of the rust fungus population. The genetic relatedness among rust fungus populations on different hosts was evaluated. Although this analysis is not a necessary step in identifying the potential for host switching of the rust pathogen, it can significantly enhance understanding of pathogen population dynamics and transmission patterns, especially in cases where there is genetic heterogeneity among pathogen populations. Through these analyses, researchers can more accurately assess the genetic background and evolutionary history of the pathogen, thus providing a scientific basis for disease management and control.

4. Line51. Taiwan is not a country. Because China has been mentioned before, it is unnecessary to list it separately. This is a very serious matter.

Author Response

  1. In the background introduction section, more discussion about the pathogen is needed, such as its taxonomic status and pathogenicity.

→ The purpose of this study is to identify the species and its life cycle. Hence, the authors believe that the introduction and discussion are appropriate, though they acknowledge that these sections could be articulated more clearly.

  1. The author also conducted inoculation experiments on a series of potential host plants to determine the host range and potential for host switching of the rust fungus.

→ The focus of this study is on elucidating the life cycle. Investigating the host range of this species is considered a future direction.

  1. Molecular marker analysis can be used to assess the genetic diversity and structure of the rust fungus population. The genetic relatedness among rust fungus populations on different hosts was evaluated. Although this analysis is not a necessary step in identifying the potential for host switching of the rust pathogen, it can significantly enhance understanding of pathogen population dynamics and transmission patterns, especially in cases where there is genetic heterogeneity among pathogen populations. Through these analyses, researchers can more accurately assess the genetic background and evolutionary history of the pathogen, thus providing a scientific basis for disease management and control.

→ The authors have conducted additional molecular analyses using the CO3 (cytochrome oxidase 3) mtDNA marker, providing valuable data for understanding population genetics.

  1. Line51. Taiwan is not a country. Because China has been mentioned before, it is unnecessary to list it separately. This is a very serious matter.

→ The authors have referred to 'China' and 'Taiwan' as geographic areas, as commonly and widely used in monographic books, without any political implication. Nevertheless, the reference to 'Taiwan' has been removed to avoid misunderstanding.

Reviewer 3 Report

Comments and Suggestions for Authors

The ms. provides interesting information on the full life cycle of the rust fungus Nyssopsora cedrelae. Authors provided evidence that the species is host alternating from Aralia to Tona using rDNA sequence data (ITS, LSU) and Cross inoculation experiments. The data are worth to be published.

References: I checked the references in l. 245 and found, that two references are missing (Cummins & Hiratsuka, the genera of rust fungi and Yadev et al. 2023 Current Research in Environmental & Applied Mycology 13: 523-549). The last reference from 2023 should be evaluated and cited here and in other parts of the ms. Citations no. 20, 22, 23 should be deleted.

Please list also sequences from GenBank in Table 1 used for Figs. 1 and 2.  

Introduction: The authors numerous times refer to the economic importance of tree rust in general and Nyssopsora cedrelae in particular. However they do not provide a single reference that documents the damage and economic  importance of this rust on Aralia/Toona. I think the whole ms. does not loose its scientific value if the phytopathological part is deleted.

In order to get sure that Aecidium araliae is not conspecific with Nyssopsora cedrelae authors have to provide a sequence of Aedium araliae on Aralia. In a former publication this fungus was considered a  species of Puccinia species (Puccinia caricis-cedralae) based on inoculation experiments. An ITS sequence, however, was not provided. This should be done in this study. 

Figure 6 is a very simple presentation of a full cycle host alternating rust. No new information is provided. Spermogonia are no spermogonia in Fig. 6. These are aecia. Delete this figure.

  Discussion: Instead of speculating I urgently recommend to order the specimen from China to check the host (Ailanthus). Or ask the curator to check to do this.  I think this is the first host-alternating species within Nyssopsora. This is a very interesting result and should be emphasized much more. Cummins & Hiratsuka and Yadav et al. do not report of host-alternating species in Nyssopsora. No host-alternating species is known for the family Nyssopsoraceae (Yadev). 

Comments on the Quality of English Language

Language poor, too many repetitions (importance of rust in intro and discussion), superfluous words (e.g. ...Korean angelica tree, is a wood plant..." - all trees are woody plants!). Numerous scientific names are wrong (in Figs. 1 and 2 Ailanthus altissimus, betheli). Recommend to give ms. to mycologist & native English speaker. 

Author Response

  1. References: I checked the references in l. 245 and found, that two references are missing (Cummins & Hiratsuka, the genera of rust fungi and Yadev et al. 2023 Current Research in Environmental & Applied Mycology 13: 523-549). The last reference from 2023 should be evaluated and cited here and in other parts of the ms. Citations no. 20, 22, 23 should be deleted.

→ Upon reviewing the references, the authors have revised the citations accordingly.

  1. Please list also sequences from GenBank in Table 1 used for Figs. 1 and 2.  

→ Reference sequences for Nyssopsora species have been included in Table 2.

  1. Introduction: The authors numerous times refer to the economic importance of tree rust in general and Nyssopsora cedrelae in particular. However they do not provide a single reference that documents the damage and economic importance of this rust on Aralia/Toona. I think the whole ms. does not loose its scientific value if the phytopathological part is deleted.

→ The authors concur with the suggestion and have revised the introduction by removing the phytopathological aspects.

  1. In order to get sure that Aecidium araliae is not conspecific with Nyssopsora cedrelae authors have to provide a sequence of Aedium araliae on Aralia. In a former publication this fungus was considered a species of Puccinia species (Puccinia caricis-cedralae) based on inoculation experiments. An ITS sequence, however, was not provided. This should be done in this study. 

→ Although acquiring a DNA sequence for Puccinia caricis-araliae poses challenges due to the preservation method of the specimens, the authors believe that morphological differences provide sufficient evidence to distinguish these species.

  1. Figure 6 is a very simple presentation of a full cycle host alternating rust. No new information is provided. Spermogonia are no spermogonia in Fig. 6. These are aecia. Delete this figure.

→ The authors have decided to retain Figure 6 to aid in understanding the life cycle of the rust, with an updated image for the spermogonia.

  1. Discussion: Instead of speculating I urgently recommend to order the specimen from China to check the host (Ailanthus). Or ask the curator to check to do this. I think this is the first host-alternating species within Nyssopsora. This is a very interesting result and should be emphasized much more. Cummins & Hiratsuka and Yadav et al. do not report of host-alternating species in Nyssopsora. No host-alternating species is known for the family Nyssopsoraceae (Yadev). 

→ In the revised discussion, speculation regarding the species in China has been omitted, and the emphasis has been placed on this being the first recorded instance of the spermogonial and aecial stages in Nyssopsoraceae.

Comments on the Quality of English Language: Language poor, too many repetitions (importance of rust in intro and discussion), superfluous words (e.g. ...Korean angelica tree, is a wood plant..." - all trees are woody plants!). Numerous scientific names are wrong (in Figs. 1 and 2 Ailanthus altissimus, betheli). Recommend to give ms. to mycologist & native English speaker. 

→ This manuscript has been reviewed and improved by a premium English language service.

Round 2

Reviewer 2 Report

Species identification based on the phylogenetic tree is inaccurate, please reanalyze.

In the ITS-LSU tree, how did the authors conduct species identification without the sequence of Nyssopsora cedrelae (ex-type)? It was inaccurate. For other referenced sequences, it should be indicated which ones are sequences of ex-type strains and which ones are not.

In the CO3 tree, the absence of other reference species of the genus Nyssopsora makes this phylogenetic tree also quite amateurish.

Author Response

  1. Species identification based on the phylogenetic tree is inaccurate, please reanalyze.

(1) In the ITS-LSU tree, how did the authors conduct species identification without the sequence of Nyssopsora cedrelae (ex-type)? It was inaccurate.

→ The ex-type sequence of Nyssopsora cedrelae is unavailable in existing databases, and the holotype specimen could not be accessed. Instead, the authors included sequences from three N. cedrelae samples collected in Japan where the type specimen was originated. The Koreana and Japanese samples were morphologically and phylogenetically consistent. The specific association of N. cedrelae with Toona sinensis and its limited East Asian distribution further support our identification.

For other referenced sequences, it should be indicated which ones are sequences of ex-type strains and which ones are not.

→ We have clearly indicated the type specimen in the phylogenetic tree (Figure 1) and Table 2, although there is only an ex-type sequences of N. alttisima.

(2) In the CO3 tree, the absence of other reference species of the genus Nyssopsora makes this phylogenetic tree also quite amateurish.

→ Currently, CO3 sequences for Nyssopsora species are lacking, and thus the present study provides an opportunity for future phylogenetic research. Our analysis showed identical sequences in all rust samples from Toona sinensis and Aralia elata, reinforcing our identification.

Reviewer 3 Report

Again, this is a very good contribution to rust research with important results. However, some major information is still missing, particularly on the life cycle. It is not uncommon that telia are formed in nature but not in cross-inoculations. But you have to mention and explain this.  

I am not an English native speaker but I notice that this ms. has not been checked by a person who is both, an English native speaker AND a mycologist. An English native speaker alone cannot do this. Maybe the editor can provide a person for this. There are too many too complicated and superfluous sentences untypical for mycological English.

see above

Author Response

  1. Too many complicated and superfluous sentences. The special finding of the work is that the authors provide evidence (cross inoculation, sequence data) that Nyssopsora cedrelae is host alternating and, consequently the first evidence for host alternating species in the Nyssopsoraceae. Especially the last sentence ("...are expected to inform future disease control strategies...) are unreasonable in my opinion. Finally, the rust is not strongly damaging Aralia and for the disease on the telial host which seems really strong (Figs. 4B) the fungus does not need an aecial host (short life cycle via urediniospores). All in all its just another heteroecious rust species and there is no need for new control strategies.

→ The authors acknowledge your points and have eliminated the unnecessary sentences, including the remarks about future disease control strategies.

Also, replace "N. asiatica develops its telial stage on A. elata." by "Nyssopsora asiatica is a microcylic autoecious species forming only telia on Aralia elata". What I am missing in the introduction and other parts of the manuscript is mentioning the whole host range of the mentioned Nyssopsora species. There are much more species than Aralia elata and T. sinensis.

→ We have clarified that Nyssopsora asiatica forms only telia on Aralia elata, making it a microcyclic autoecious species.

Whenever overview articles are cited only Cummins & Hiratsuka or Lohsomboon et al. are cited. Obviously, the authors don't know about the most detailed study on Nyssopsora by the Dutch author Lütjeharms (1937). Lütjeharms, W. J. (1937). Vermischte Mykologische Notizen I. Blumea. Supplement, 1(1), 142–161.(available on the internet, by the way). The authors have to study this article and cite it.

→ Additional host plants beyond Aralia elata and Toona sinensis have been incorporated into our discussion, with appropriate references added.

  1. The work could be much better and stronger if the authors would have sequenced Nyssopsora asiatica as well. This species seems closest to N. cedrelae according to Tranzschel's law.

→ As there is no report of N. asiatica in Korea, the differentiation between N. cedrelae and N. asiatica is based on comparison with reference sequences from China available in GenBank.

Material: Abbreviations of public herbaria are listed in Index Herbariorum. I couldn't find KSNUH and KUS-F indeed seems to be simply KUS. Please clarify. Also explain abbreviations in Table 1. Table 1: This is not only a list of Nyssopsoraceae but contains also species of other families.

→ ‘KUS’ is an official herbarium in Index Herbariorum, and we added “-F” for fungal specimens. ‘KSNUH’ means Kunsan National University Herbarium, which is not yet registered in Index Herbariorum. We explained abbreviations for all herbarium specimens in Tables 1 and 2.

  1. 3.3. Life cycle I am missing the information on telia and basidia formed on Toona. Please describe whether telia were formed or not. If not discuss this.

→ We did not observe telial and basidial stages in a pathogenicity test. Authors explained this in text.

Table 3: delete Lohsmboon, Kakishima and Hori. The biometric data of Lütjeharms (1937) should be listed here as well.

→ Following a careful review of Lütjeharms (1937), the authors incorporated the available morphological data into Table 3. We prefer to keep the measurement of Lohsmboon, Kakishima and Hori.

Fig. 6: Again, this figure is superfluous and does not provide any new information.

→ We deleted Figure 6.

  1. As mentioned above the following article is missing: Lütjeharms, W. J. (1937). Vermischte Mykologische Notizen I. Blumea. Supplement, 1(1), 142–161.

→ The paper (Lütjeharms, 1937) has been added to the manuscript.

Major comments. However, some major information is still missing, particularly on the life cycle. It is not uncommon that telia are formed in nature but not in cross-inoculations. But you have to mention and explain this.  

→ We mentioned it in 3.3.

I am not an English native speaker but I notice that this ms. has not been checked by a person who is both, an English native speaker AND a mycologist. An English native speaker alone cannot do this. Maybe the editor can provide a person for this. There are too many too complicated and superfluous sentences untypical for mycological English.

→ Addressing the concerns regarding the manuscript's language quality, all authors have carefully reviewed the text for English language accuracy, with a focus on clarity and mycological terminology. We believe this revision has significantly improved the manuscript's readability and scientific information.

Round 3

Reviewer 2 Report

The authors of this manuscript carried out analyses of Nyssopsora cedrelae on Toona sinensis and Aralia elata in Korea, using methods that included phylogenetic, morphological, and pathogenic evaluations. The manuscript is well-prepaired and adequately presented, and the methods used were reliable. The results presented in the manuscript are convincing and adequately justified. Minor revisions were suggested before publication.

The authors mentioned briefly that no teliospores were observed in the pathogenicity test, but they did not explain why. This is a crucial detail that requires clarification.

In the pathogenicity test, it is also suggested that authors attempt to infect Aralia elata with aeciospores from Toona sinensis. This approach may potentially lead to discoveries.

Author Response

The authors mentioned briefly that no teliospores were observed in the pathogenicity test, but they did not explain why. This is a crucial detail that requires clarification. 

--> The authors acknowledge the reviewer's comment regarding the absence of the telial stage in our pathogenicity tests. Currently, the specific reasons behind the lack of teliospore production in our study are not known to us. The production of teliospores, resulting from the change from the uredinial to the telial stage in the life cycle of rust pathogens, is believed to be influenced by a complex interplay of biotic factors related to the host plant and environmental conditions. Unfortunately, such prerequisites are beyond the scope of our current understanding and were not directly assessed in this study.

In the pathogenicity test, it is also suggested that authors attempt to infect Aralia elata with aeciospores from Toona sinensis. This approach may potentially lead to discoveries.

→ The suggestion for reciprocal inoculation to complete the life cycle of the pathogen, specifically by attempting to infect Aralia elata with teliospores and basidiospores from Toona sinensis, is appreciated for its potential to yield significant insights. However, the practicality of performing inoculations with aeciospores and teliospores from the respective host plants is a challenge, primarily due to the difficulty in obtaining viable samples. Furthermore, it would require a time frame extending beyond one year. Consequently, our study employed a one-way inoculation approach to substantiate the life cycle of this species.